# Associations of frailty with partial and absolute sedentary behaviours among older adults: A STROBE-compliant analysis of modifiability by gender and age

**Nestor Asiamah** [1,2]*, **Hafiz T. A. Khan**[3], **Cosmos Yarfi**[4], **Simon Mawulorm Agyemang**[5], **Reginald Arthur-Mensah Jnr**[6], **Faith Muhonja**[7], **Sarra Sghaier**[2], **Kyriakos Kouveliotis**[8]

1 Division of Interdisciplinary Research and Practice, School of Health and Social Care, University of Essex, Colchester, United Kingdom, 2 Department of Geriatrics and Gerontology, Africa Centre for Epidemiology, Accra, Ghana, 3 College of Nursing, Midwifery, and Healthcare, University of West London, London, United Kingdom, 4 Department of Physiotherapy and Rehabilitation Sciences, University of Health and Allied Sciences, Ho, Ghana, 5 Department of Science/Health, Physical Education and Sports, Abetifi College of Education, Abetifi, Ghana, 6 Department of Nursing and Midwifery, Pentecost University, Accra, Ghana, 7 Department of Community Health, Amref International University, Nairobi, Kenya, 8 Berlin School of Business and Innovation, Berlin, Germany

* n.asiamah@essex.ac.uk

**Data Availability Statement:** All relevant data are within the paper and its Supporting information files.

## Abstract

### Background

Research shows that frailty is associated with higher sedentary behaviour, but the evidence to date regarding this association is inconclusive. This study assessed whether the above association is moderated or modified by gender and age, with sedentary behaviour measured with a more inclusive method.

### Methods

This study adopted a STROBE-compliant cross-sectional design with sensitivity analyses and measures against common methods bias. The participants were community-dwelling older adults (mean age = 66 years) in two Ghanaian towns. A self-reported questionnaire was used to collect data from 1005 participants after the minimum sample size necessary was calculated. The hierarchical linear regression analysis was used to analyse the data.

### Results

After adjusting for the ultimate confounders, frailty was associated with higher sedentary behaviour (β = 0.14; t = 2.93; p <0.05) as well as partial and absolute sedentary behaviour. Gender modified the above associations in the sense that frailty was more strongly associated with sedentary behaviour among women, compared with men. Age also modified the association between frailty and sedentary behaviour, which suggests that frailty was more strongly associated with higher sedentary behaviour at a higher age.

**Funding:** The authors received no funding for this study.

**Competing interests:** The authors declare no competing interests.

## Conclusion

Sedentary behaviour could be higher at higher frailty among older adults. Frailty is more strongly associated with sedentary behaviour at a higher age and among women, compared with men.

## Introduction

Physical activity [PA] can protect the individual against chronic conditions such as diabetes, stroke, and hypertension as well as early mortality [1–4]. On the flip side, sedentary behaviour increases the risk of the above conditions [2, 5, 6]. Sedentary behaviour has been defined as a waking behaviour that requires an energy expenditure of not more than 1.5 basal metabolic rates [7]. It is simply defined as too much sitting [7]. A condition that makes it less possible for older adults to avoid sedentary behaviour is frailty [8], which is the condition of being weak and delicate [8, 9]. Though frailty has been defined in different ways in different contexts, we employed this definition in the context of clinical practice to adopt a clinical measure and report implications for clinical practice. Research [8, 10, 11] has shown that sedentary behaviour increases with frailty. Though the literature also recognises frailty as an outcome of sedentary behaviour [10, 12, 13], there is a consensus among researchers that frail people are unlikely to avoid sedentary behaviour since they generally lack physical functional capacity [8, 14–16].

A systematic review [10] reveals that though some studies have assessed the association between frailty and sedentary behaviour, research on this relationship is inconclusive. This review suggests that measures used in previous research possibly underestimated sedentary behaviour since these measures characterised a single item asking individuals to report their sedentary time. A single item is unlikely to capture sedentary times spent in various situations (e.g., reclining, driving, viewing television), especially in older adults who may be unable to recall these times and report them on a single question. This concern is echoed by some researchers [2] who attempted to provide a more inclusive measure of sedentary behaviour. This study builds on the foregoing research by showing differences in two types of sedentary behaviour (i.e., partial and absolute) in terms of their association with frailty, age, and gender. This analysis may improve researchers' understanding of why research must be cognisant of potential differences in the intensities of different types of sedentary behaviour. It is expected to be an improvement over traditional subjective measures assuming that sedentary behaviours are of the same intensity.

Though objective measures are recognised as the ultimate for measuring sedentary behaviour [8, 10], they do not allow people to relate their lived experiences, are generally expensive, and can easily be damaged [10]. This view suggests that many researchers may be unable to buy and use objective measures. More so, subjective measures or questionnaires are needed to measure actual views and experiences regarding sedentary behaviour, especially in contexts where objective measures are not feasible or suitable. This study is, therefore, important as it provides a potentially enhanced subjective measure that can be used in the foregoing contexts. It is based on a recent recommendation [2] for studies utilising a more inclusive or multi-item measure.

There is a paucity of research assessing potential modifiers of the association between frailty and sedentary behaviour. Gender and age, for example, are pivotal variables in the gerontological literature that can explain differences in frailty [17–21], but only two studies [22, 23], both of which were undertaken in developed non-African countries, had assessed gender differences in the above association. No study has assessed age as a potential modifier of the above

association, though this potential role of age has implications for theory and practice. This study, therefore, for the first time investigated whether gender and age can modify the association of frailty with sedentary behaviour measured with multiple questions and two domains, instead of a single item. An attempt was made to answer the following research questions: *(1) is there an association between frailty and sedentary behaviour; (2) does gender modify the association between frailty and sedentary behaviour*, and *(3) does age modify the association between frailty and sedentary behaviour*?

## Methods

### Design

This study adopted a STROBE-compliant cross-sectional design that is summarised in Fig 1.

### Participants, sample, and recruitment

The participants were community-dwelling older adults aged 50 years or higher in Ghana. We targeted older adults in two peri-urban towns, each with a combination of low and high socio-

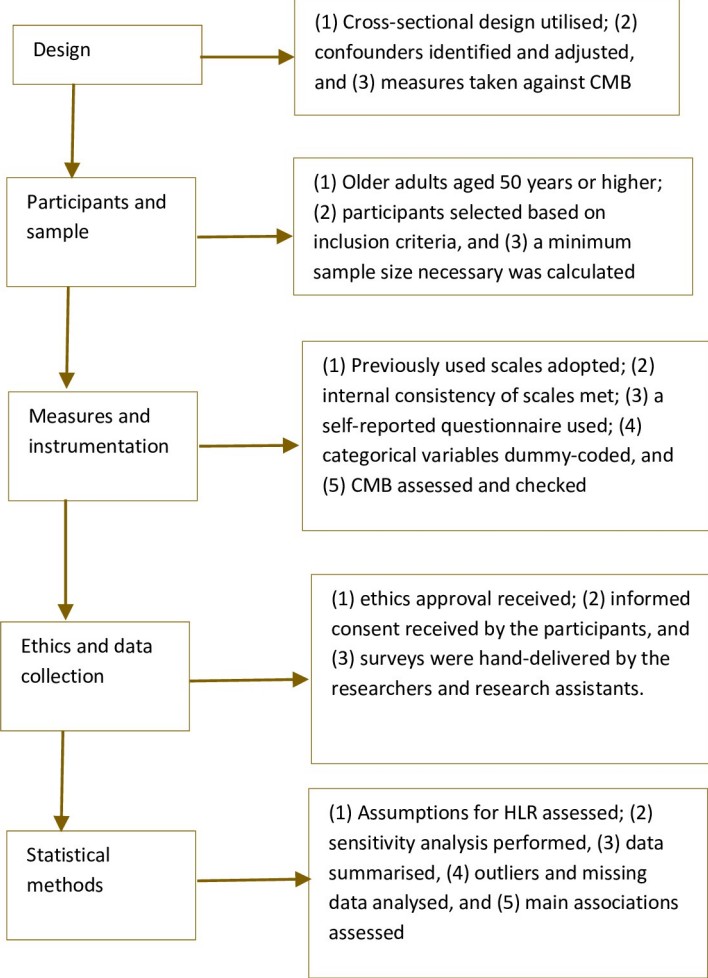

**Fig 1. A Flowchart of the STROBE-compliant design.** Note: CMB–common methods bias, HLR–hierarchical linear regression.

economic neighbourhoods. The following inclusion criteria were used to select the participants: (1) having at least a basic education qualification (i.e., basic school living certificate), which we used as an indicator of the ability to complete the questionnaire in English; (2) being a permanent community-dwelling resident aged 50 years or higher, and (3) the ability to walk independently for at least 10 minutes [24]. We calculated the minimum sample size necessary for this study with the G*Power software and recommended statistics (i.e., effect size = 0.2, α = 0.05, power = 0.8) suited for our sample [25]. The minimum sample estimated for HLR analysis with a maximum of 10 predictors was 91. To maximise the statistical power of our tests, we selected as many eligible individuals as possible. There was no sampling frame for this study, so potential participants were selected by research assistants at community and social centres such as supermarkets and events. Research assistants interviewed potential participants at various community centres to assess their eligibility. A total of 1039 older adults who met the above criteria were selected.

## Measures and operationalization

We followed a recent study [2] to measure sedentary behaviour as a construct of partial sedentary behaviour and absolute sedentary behaviour. Partial sedentary behaviour is time (in minutes) spent sitting while doing an activity such as playing a game or driving on a typical day. It was measured by asking the participants to report time spent sitting in six situations. Absolute sedentary behaviour is the time (in minutes) spent on a typical day while sitting without physically moving any part of the body. It was measured by asking the participants to report time spent while sitting in six situations (e.g., lying down, reclining, viewing TV). Sedentary behaviour was the sum of time reported on the 12 items whereas partial and absolute sedentary behaviours were the sum of their respective 6 items. Appendix A in S1 Appendix shows questions and items used to measure sedentary behaviour. Frailty was measured with the 15-item Tilburg Clinical Frailty Index (see Appendix B in S1 Appendix) with two descriptive anchors (yes–1; no–0) adopted in whole from a previous study [26]. As noted earlier, we used this clinical measure of frailty to be able to identify implications for clinical or geriatric practice. This scale produced a satisfactory internal consistency in the form of Cronbach's alpha coefficient = 0.92.

Self-reported health, chronic disease status (CDS), income, age, gender, education, walkability, and social network size are recognised in the literature as potential predictors of frailty [27–29] that can confound the primary relationships tested. These variables were measured based on methods previously used [2, 24, 28]. Walkability was measured with the 11-item Australian version of the Neighbourhood Environment Walkability Scale (NEWS-AU). This tool has five anchors (i.e., strongly disagree–1, disagree–2, somewhat agree–3, agree–4, and strongly agree–5) and produced a satisfactory Cronbach's alpha = 0.78. This scale was used because it produced satisfactory results, including psychometric properties, in a previous study focused on an older Ghanaian sample [24]. The data on walkability and frailty were generated by adding up their items using the compute function in SPSS [24]. Appendix C in S1 Appendix shows items used to measure walkability.

Social network size was measured as a discrete variable by asking the participants to report the current number of close social ties (e.g., friends, workmates, blood relations, and neighbours) they had performed social and physical activities with over the past 7 days (see Appendix D in S1 Appendix). Income was measured as a continuous variable by asking the participants to report their current gross monthly income in Ghana cedis. Context experience was measured as a discrete variable by asking the participants to report how long [in years] they had lived in their current neighbourhood. Relationship status (i.e., married or in a

relationship–1 and not married or in a relationship–0), CDS (i.e., none–0, and one or more–1], gender [male–0 and female–1), and self-reported health (i.e., poor–0 and good–1) were measured as categorical variables, which were coded into dummy-type variables for regression modelling. Age was measured as a discrete variable by asking the participants to report their age. Education was measured as a continuous variable by asking the participants to report their years of schooling.

## Instrumentation, ethics, and data collection

A self-reported questionnaire was utilised to collect the data. The first section of the questionnaire presented personal and demographic information, most of which were the potential confounders. The second and third sections captured measures of frailty and walkability respectively whereas the final section presented questions measuring sedentary behaviour. The questionnaire had a preamble containing the study aim, benefits, a statement emphasizing ethical considerations, and instructions for completing the survey. Two steps taken in previous studies [24, 30] were taken to minimise or avoid common methods bias (CMB). Appendix E in S1 Appendix shows specific steps taken to assess CMB and confirm its absence in the data.

Before data were collected, the participants were informed about the purpose, benefits, risks, and confidentiality of the information they provided through an informed consent sheet. Through a participant information sheet, the participants provided a written informed consent before participating in the study. The participants received information about how their information would be anonymised and stored and how long the information would be stored. The recruitment of the participants was carried out between 1st October and 3rd November 2022. No minors such as children participated in this study.

Two of the researchers (CY and SA) supported by research assistants coordinated data collection in two communities where questionnaires were hand-delivered to the participants. The coordinators arranged for the completed questionnaires to be returned instantly or after two weeks, depending on what worked for the participants. Data were collected over four weeks between October and November 2022. Out of 1039 questions administered and returned, 34 were discarded because they were completed halfway or were not completed at all. Thus, 1005 were analysed.

## Statistical analyses

The data were analysed in two stages with SPSS 28 (IBM SPSS Inc., New York). In the first stage, descriptive statistics were used to summarise the data, the relevant assumptions governing the use of HLR analysis were assessed, and the first sensitivity analysis was performed. Following previous research [24, 31], five assumptions governing the use of HLR analysis were assessed. Appendix F in S1 Appendix shows all the steps taken to assess and meet these assumptions. The final aspect of the exploratory analysis was the first sensitivity analysis to screen for the ultimate confounding variables, which followed a recent approach [24]. This analysis was utilised to select only variables more likely to confound the primary relationships. Appendix G in S1 Appendix shows the steps taken in this analysis. The ultimate confounders according to this analysis were CDS and context experience.

In the second stage, we tested the hypothesized relationships shown in Fig 2. Pearson's correlation was used to compute bivariate correlations between relevant variables. Six regression models were then fitted; 3 of these were baseline (non-adjusted) models whereas the other 3 were adjusted or ultimate models on which this study's conclusions are based. The first of the non-adjusted models assessed the association between frailty and sedentary behaviour ($H_1$) as well as its two domains (i.e., partial and absolute sedentary behaviours) without the ultimate

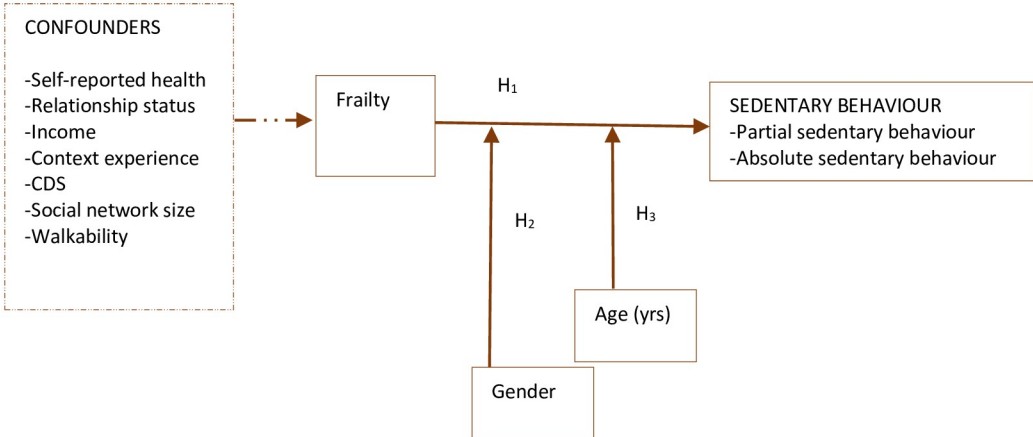

**Fig 2. The association between frailty, sedentary behaviour, age, and gender. Note**: Broken arrows represent potential confounding; CDS–chronic disease status; PAS–partial sedentary behaviour; ASB–absolute sedentary behaviour; $H_1$ – frailty is associated with sedentary behaviour; $H_2$ –gender moderates the association between frailty and sedentary behaviour; $H_3$ –age moderates the association between frailty and sedentary behaviour.

confounders. The second non-adjusted model assessed the association between sedentary behaviour and the interaction of frailty with gender ($H_2$). The third non-adjusted model tested the association between sedentary behaviour and the interaction of frailty with age ($H_3$). Models 4, 5 and 6 were built on models 1, 2 and 3 respectively by infusing the ultimate confounding variables.

We assessed pure moderation [24], which means we were interested only in whether the strength of the association between frailty and sedentary behaviour was significantly modified by gender and age. To assess this type of moderation, we computed the two interaction terms (i.e., genderXfrailty and ageXfrailty) and evaluated their association with sedentary behaviour and its domains. GenderXfrailty was the interaction between frailty and gender whereas AgeXfrailty was the interaction between frailty and age. We detected the statistical significance of the tests at p<0.05.

## Results

### Summary statistics

Table 1 shows summary statistics on all variables. Nearly 50% (n = 500) of the participants were men whereas the average age of the participants was about 66 years (Mean = 66.42, SD = 10.54). The average frailty level was 22 (Mean = 21.22, SD = 2.03) and the average sedentary behaviour was 1795 minutes (Mean = 1795.02; SD = 1440.10). Table 2 shows Pearson's correlations between relevant variables. Frailty was positively correlated with sedentary behaviour (r = 0.202; p<0.001; two-tailed) and its two domains, which means that high sedentary behaviour was associated with higher frailty. Age (r = 0.145; p<0.001; two-tailed) and gender (r = 0.186; p<0.001; two-tailed) were also positively correlated with frailty, which formed the basis of their potential moderation of the association between frailty and sedentary behaviour.

### Key findings

Table 3 shows regression results corresponding to the above correlations. In the first ultimate model (model 2), frailty was associated with sedentary behaviour (β = 0.14; t = 2.93; p<0.05)

**Table 1. Summary statistics on all variables included in the study (n = 1005).**

| Variables | Group | Frequency/Mean | Percent(%)/SD |
|---|---|---|---|
| | | Categorical variables | |
| Gender | Men | 500 | 49.75 |
| | Women | 505 | 50.25 |
| Chronic disease status | None | 355 | 35.32 |
| | $\geq 1$ | 645 | 64.18 |
| | Missing | 5 | 0.50 |
| Relationship status | Not in a relationship | 245 | 24.38 |
| | In a relationship | 760 | 75.62 |
| Self-reported health | Poor | 340 | 33.83 |
| | Good | 655 | 65.17 |
| | Total | 1005 | 100.00 |
| | | Continuous variables | |
| Income (₵) | - - - | 782.88 | 930.42 |
| Age (yrs) | - - - | 66.42 | 10.54 |
| Context experience (yrs) | - - - | 34.43 | 24.88 |
| Education (yrs) | - - - | 12.21 | 3.21 |
| Frailty | - - - | 21.22 | 2.03 |
| Partial sedentary behaviour | - - - | 762.67 | 675.78 |
| Absolute sedentary behaviour | - - - | 1032.36 | 1086.78 |
| Sedentary behaviour | - - - | 1795.02 | 1440.10 |

Note: - - - Not applicable; SD–standard deviation; 'Mean' and SD apply to continuous variables whereas frequency and % apply to categorical variables.

and each of its domains. Thus, higher frailty was associated with higher sedentary behaviour and its domains. In the second ultimate model (model 4), the interaction between gender and frailty was positively associated with partial sedentary behaviour ($\beta = 0.11$; t = 2.36; p<0.05) but not absolute sedentary behaviour and sedentary behaviour. The coefficient in model 2 has reduced from 0.14 to 0.11 (a 27% change in $\beta$) for sedentary behaviour and to -0.03 and 0.03 for partial and absolute sedentary behaviour respectively (a 367% change in $\beta$). Gender weakened the association between frailty and sedentary behaviour, which means that frailty was

**Table 2. Pearson's bivariate correlations between relevant variables (n = 1005).**

| Variable | 1 | 2 | 3 | 4 | 5 | 6 | 7 | 8 |
|---|---|---|---|---|---|---|---|---|
| 1. Frailty | 1 | .202** | .230** | .269** | .186** | .149** | .145** | .105** |
| 2. Partial sedentary behaviour | | 1 | .297** | .693** | .106* | 0.077 | .232** | .333** |
| 3. Absolute sedentary behaviour | | | 1 | .894** | -0.049 | .230** | .377** | .247** |
| 4. Sedentary behaviour | | | | 1 | 0.013 | .210** | .394** | .342** |
| 5. Gender (ref.–men) | | | | | 1 | 0.01 | 0.056 | -0.047 |
| 6. CDS (ref.–none) | | | | | | 1 | .378** | .280** |
| 7. Age (yrs) | | | | | | | 1 | .545** |
| 8. Context experience (yrs) | | | | | | | | 1 |

**p<0.001;

*p<0.05;

CDS–chronic disease status

**Table 3. The associations between frailty, sedentary behaviour, gender, and age (n = 1005).**

| Model | Predictor | Partial sedentary behaviour (mins) | | | | Absolute sedentary behaviour (mins) | | | | Sedentary behaviour (mins) | | | |
|---|---|---|---|---|---|---|---|---|---|---|---|---|---|
| | | B | SE | β(t) | 95% CI | B | SE | β(t) | 95% CI | B | SE | β(t) | 95% CI |
| 1 | (Constant) | -559.36 | 308.97 | (-1.81) | ±1214.55 | -1382.42 | 493.801 | (-2.8)* | ±1941.088 | -1941.78 | 647.662 | (-3.00)* | ±2545.904 |
| | Frailty | 61.24 | 14.24 | 0.20(4.30)** | ±55.97 | 111.867 | 22.755 | 0.23(4.92)** | ±89.446 | 173.111 | 29.845 | 0.27(5.80)** | ±117.316 |
| 2 | (Constant) | -373.62 | 299.73 | (-1.25) | ±1178.26 | -1069.95 | 482.687 | (-2.22)* | ±1897.488 | -1443.57 | 621.196 | (-2.32)* | ±2441.982 |
| | Frailty | 41.73 | 14.23 | 0.14(2.93)* | ±55.95 | 79.078 | 22.922 | 0.16(3.45)** | ±90.107 | 120.803 | 29.499 | 0.19(4.10)** | ±115.964 |
| | CDS (ref.–none) | 11.2 | 63.02 | 0.01(0.18) | ±247.75 | 374.233 | 101.494 | 0.17(3.69)** | ±398.982 | 385.431 | 130.618 | 0.13(2.95)* | ±513.472 |
| | Context experience (yrs) | 11.31 | 1.72 | 0.31(6.58)** | ±6.76 | 11.164 | 2.77 | 0.19(4.03)** | ±10.89 | 22.475 | 3.565 | 0.29(6.31)** | ±14.014 |
| 3 | (Constant) | 438.11 | 101.24 | (4.33)** | ±397.97 | 946.806 | 164.893 | (5.74)** | ±648.179 | 1384.917 | 217.587 | (6.37)** | ±855.315 |
| | GenderXfrailty | 9.3 | 2.75 | 0.16(3.38)** | ±10.82 | 2.452 | 4.484 | 0.03(0.55) | ±17.627 | 11.756 | 5.917 | 0.10(1.99)* | ±23.26 |
| 4 | (Constant) | 290.77 | 99.81 | (2.91)* | ±392.36 | 670.831 | 162.296 | (4.13)** | ±638.002 | 961.603 | 210.08 | (4.58)** | ±825.846 |
| | GenderXfrailty | 6.4 | 2.71 | 0.11(2.36)* | ±10.64 | -3.216 | 4.403 | -0.03(-0.73) | ±17.308 | 3.183 | 5.699 | 0.03(0.56) | ±22.404 |
| | CDS (ref.–none) | 21.76 | 62.95 | 0.02(0.35) | ±247.45 | 449.64 | 102.358 | 0.21(4.39)** | ±402.378 | 471.401 | 132.494 | 0.16(3.56)** | ±520.848 |
| | Context experience (yrs) | 11.71 | 1.71 | 0.32(6.85)** | ±6.73 | 13.171 | 2.783 | 0.22(4.73)** | ±10.938 | 24.884 | 3.602 | 0.32(6.91)** | ±14.159 |
| 5 | (Constant) | -146.32 | 142.8 | (-1.03) | ±561.32 | -1025.31 | 218.3 | (-4.70)** | ±858.12 | -1171.63 | 283.49 | (-4.13)** | ±1114.37 |
| | AgeXFrailty | 0.68 | 0.11 | 0.30(6.52)** | ±0.41 | 1.54 | 0.16 | 0.42(9.66)** | ±0.63 | 2.23 | 0.21 | 0.46(10.72)** | ±0.82 |
| 6 | (Constant) | -26.13 | 143.82 | (-0.18) | ±565.35 | -1002.62 | 221.96 | (-4.52)** | ±872.56 | -1028.75 | 286.17 | (-3.60)** | ±1124.95 |
| | AgeXFrailty | 0.46 | 0.12 | 0.20(3.86)** | ±0.47 | 1.39 | 0.18 | 0.38(7.59)** | ±0.72 | 1.85 | 0.24 | 0.38(7.83)** | ±0.93 |
| | CDS (ref.–none) | -22.64 | 63.96 | -0.02(-0.35) | ±251.44 | 233.02 | 98.72 | 0.11(2.36)* | ±388.08 | 210.38 | 127.28 | 0.07(1.65) | ±500.33 |
| | Context experience (yrs) | 9.38 | 1.83 | 0.25(5.12)** | ±7.21 | 4.26 | 2.83 | 0.07(1.50) | ±11.13 | 13.64 | 3.65 | 0.17(3.74)** | ±14.35 |

**p<0.001;

*p<0.05;

SE–standard error; CI–confidence interval (of B); CDS–chronic disease status; Tolerance for each predictor ≥0.5; Durbin-Watson for each model is approximately 2; total adjusted $R^2$ ranged from 0.1%-2% for the simple models (with only 1 predictor) and 2%-25% for the multiple models (with two or more predictors). The F-tests of the simple models were significant at p<0.05 whereas the F-tests of the multiple models were significant at p<0.001.

more strongly associated with sedentary behaviour and its two domains among women, compared with men.

In model 6, the interaction between age and frailty was positively associated with sedentary behaviour (β = 0.20; t = 3.86; p<0.001) and its two domains. The coefficient in model 2 has increased from 0.14 to 0.20 (a 43% change in β), 0.38 (a 271% change in β), and 0.38 (a 271% change in β) for partial sedentary behaviour, absolute sedentary behaviour, and overall sedentary behaviour respectively. This result suggests that frailty was more strongly associated with sedentary behaviour and its domains at a higher age. The non-adjusted models (i.e., models 1, 3, and 5) produced regression weights of different sizes compared to the ultimate models, suggesting that the ultimate confounding variables influenced the primary relationships tested.

## Discussion

This study aimed to assess the association of frailty with sedentary behaviour and its two domains, namely partial and absolute sedentary behaviours. A positive association between frailty and sedentary behaviour and its two domains was found, which implies that sedentary behaviour was higher at higher frailty. Older people experiencing higher frailty could report

higher sedentary time. This result is consistent with studies undertaken in various countries [11, 32, 33]. A cross-sectional study in Brazil, to be specific, found a positive association between frailty and sedentary behaviour [32]. This evidence has been confirmed in studies carried out in Canada [11] and Spain [33]. Noteworthy is the fact that these and other previous studies utilized a single item to measure sedentary behaviour. Our method captured sitting in situations older adults often find themselves in [2] and, thus, measured sedentary behaviours more inclusively. Because different effect sizes were found between frailty and the two domains of sedentary behaviour, each of the domains constitutes a unique component of sedentary behaviour. As reasoned by some researchers [2], therefore, the two domains may have different impacts on individual health.

The foregoing result can be explained by the Disengagement Theory of Ageing [DTA] [34], which assumes that PA reduces in the ageing process due to the individual's disengagement with society. It adds that disengagement with society is the consequence of factors including a gradual decline in physical functional ability. Frailty is the consequence of a decline in physical function [26, 33], which means that higher sedentary behaviour may be associated with frailty. Some previous studies [11, 32, 33] have explained the positive relationship between frailty and sedentary behaviour with a similar line of argument. An important practical implication is that enabling ageing individuals to take steps to maintain their physical function or avoid early onset of frailty can buffer sedentary behaviour. This being so, interventions to discourage sedentary behaviour, including those rolled out in healthcare, could aim to prevent frailty.

The significant positive association between frailty and sedentary behaviour was moderated or weakened by gender, which means that this association was stronger among women. This evidence stems from the association between frailty and gender, which has been confirmed in some studies [17–19]. Drawing on the above adaptation of the DTA, groups having different frailty levels would have different levels of susceptibility to sedentary behaviour. Moreover, men and women have different health profiles and age in different ways [18], so frailty can be expected to differently influence men and women regarding their sedentary behaviour. As stated earlier, though, this study was the first to empirically evidence the modifiability of the association between frailty and sedentary behaviour by gender. Possibly, the association between frailty and sedentary behaviour as well as other relationships implied by the DTA could be modified by gender since inequalities between men and women in the ageing process have been reported [17, 18, 34]. A practical implication of our result is that susceptibility to sedentary behaviour as a risk factor due to frailty differs between men and women. As such, interventions preventing or reducing frailty and its potential influence on sedentary behaviours must consider this difference and prioritise those at a higher risk.

The significant positive association of frailty with sedentary behaviour was moderated or strengthened by age, which means that this relationship was stronger at a higher age. Similarly, frailty more strongly predicted sedentary behaviour among older adults at a higher chronological age. This result is rooted in the association between age and frailty, which has been confirmed by some studies [20, 21, 28]. A systematic review [21], for example, reported that frailty increases with age, which means the impacts of frailty on health and its risks (e.g., sedentary behaviour) may differ with age. This deduction is corroborated by the DTA, which implies that social disengagement and sitting can be due to frailty and would increase over time. This study, in effect, may affirm the reasoning that sedentary behaviour is an age-related risk or a consequence of ageing. Given the role of gender found in this study, it is possible that other personal factors that are age-related can modify the association between frailty and sedentary behaviour. Future studies incorporating more personal modifiers may, thus, advance our study.

This study has some limitations, the first of which is its cross-sectional design which is unable to establish causation. A cross-sectional design, nonetheless, is a useful method that could inform practice and set the basis for designs (e.g., experimental) that can establish causation [24, 35]. This study also employed a non-probability sampling method to select study participants. As such, its findings may have limited generalisability. Since cognitive impairment is generally higher among older adults, this could have affected participant ratings, and we could not assess the cognitive ability of the participants to ensure we included only those with sufficient cognitive ability. We could not use an objective measurement method (e.g., accelerometer) due to limited funding available for this study. Therefore, we encourage future researchers to combine objective and subjective measurement methods and compare their results. Our measurement was susceptible to recall bias since we utilised subjective measures. We tried to avoid or minimise this issue with measures against CMB and by asking older adults to report activities performed over the last 7 days.

This study has several strengths. This study was the first to assess the association of frailty with sedentary behaviour measured with multiple items representing sitting behaviour performed in different situations. This approach is an improvement over the single-item method, which can under-estimate sedentary behaviour [10]. This study followed the STROBE checklist and, thus, serves as a model for future studies, given that most cross-sectional studies in gerontology do not follow this or related checklists [36]. Appendix H in S1 Appendix shows recommendations of the STROBE met. Our design included sensitivity analyses previously applied [24, 36] to minimise confounding. This study would have reported wrong effect sizes or associations if we failed to adjust for the ultimate confounders. Suffice it to say that this study employed a robust cross-sectional design that could guide future research.

## Conclusion

Sedentary behaviour could be higher at higher frailty levels among older adults, and frailty was more strongly associated with higher sedentary behaviour among women. Frailty is more strongly associated with sedentary behaviour at a higher age. Interventions aimed at reducing frailty among older adults could buffer sedentary behaviour. Personal factors could modify the association between frailty and sedentary behaviour, so future studies may investigate whether other factors [e.g., income] modify the association between frailty and sedentary behaviour.

## Supporting information

**S1 Appendix.**
(ZIP)

**S1 Data.**
(SAV)

## Acknowledgments

We thank Mr Richard Eduafo for supporting us to collect data. This manuscript is a product of the Nestor Asiamah Research Mentorship (NARM) programme, and we thank all stakeholders who have made this programme successful.

## Author Contributions

**Conceptualization:** Nestor Asiamah, Hafiz T. A. Khan, Cosmos Yarfi, Simon Mawulorm Agyemang, Reginald Arthur-Mensah Jnr, Faith Muhonja, Sarra Sghaier.

**Data curation:** Nestor Asiamah, Cosmos Yarfi.

**Formal analysis:** Nestor Asiamah.

**Funding acquisition:** Simon Mawulorm Agyemang, Reginald Arthur-Mensah Jnr.

**Investigation:** Cosmos Yarfi, Reginald Arthur-Mensah Jnr.

**Methodology:** Nestor Asiamah, Hafiz T. A. Khan, Simon Mawulorm Agyemang, Reginald Arthur-Mensah Jnr.

**Resources:** Cosmos Yarfi, Simon Mawulorm Agyemang, Reginald Arthur-Mensah Jnr.

**Supervision:** Nestor Asiamah, Hafiz T. A. Khan, Kyriakos Kouveliotis.

**Validation:** Nestor Asiamah, Hafiz T. A. Khan, Cosmos Yarfi, Reginald Arthur-Mensah Jnr, Faith Muhonja, Sarra Sghaier, Kyriakos Kouveliotis.

**Visualization:** Nestor Asiamah, Hafiz T. A. Khan, Simon Mawulorm Agyemang, Faith Muhonja, Sarra Sghaier, Kyriakos Kouveliotis.

**Writing – original draft:** Nestor Asiamah, Hafiz T. A. Khan.

**Writing – review & editing:** Nestor Asiamah, Hafiz T. A. Khan, Cosmos Yarfi, Simon Mawulorm Agyemang, Reginald Arthur-Mensah Jnr, Faith Muhonja, Sarra Sghaier, Kyriakos Kouveliotis.

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
