## [Decision Letter · Decision Letter 0]

14 Sep 2023

PONE-D-23-22789Associations of Frailty with Partial and Absolute Sedentary Behaviours among Older Adults: A STROBE-Compliant Analysis of Modifiability by Gender and AgePLOS ONE

Dear Dr. Asiamah,

Thank you for submitting your manuscript to PLOS ONE. After careful consideration, we feel that it has merit but does not fully meet PLOS ONE’s publication criteria as it currently stands. Therefore, we invite you to submit a revised version of the manuscript that addresses the points raised during the review process.

We look forward to receiving your revised manuscript.

Kind regards,

Kiyoshi Sanada, PhD

Academic Editor

PLOS ONE

Journal Requirements:

4. We are unable to open your Supporting Information file DATA file.sav. Please kindly revise as necessary and re-upload.

Reviewers' comments:

Reviewer's Responses to Questions

**Comments to the Author**

1. Is the manuscript technically sound, and do the data support the conclusions?

Reviewer #1: Yes

Reviewer #2: Yes

2. Has the statistical analysis been performed appropriately and rigorously? 

Reviewer #1: Yes

Reviewer #2: No

3. Have the authors made all data underlying the findings in their manuscript fully available?

Reviewer #1: Yes

Reviewer #2: Yes

4. Is the manuscript presented in an intelligible fashion and written in standard English?

Reviewer #1: Yes

Reviewer #2: Yes

5. Review Comments to the Author

Reviewer #1: This study is excellent in that it follows the STROBE checklist and confounding factors are also done in detail.

It was a good learning experience for me as well as a reference for conducting my own cross-sectional study.

I would like to ask you a few questions and would be happy if you could answer them.

1.[We followed a recent study [2] to measure sedentary behaviour as a construct of partial sedentary behaviour and absolute sedentary behaviour. Partial sedentary behaviour is time (in minutes) spent sitting while doing an activity such as playing a game or driving on a typical day. It was measured by asking the participants to report time spent sitting in six situations. Absolute sedentary behaviour is the time (in minutes) spent on a typical day while sitting without physically moving any part of the body. It was measured by asking the participants to report time spent while sitting in six situations (e.g., lying down, reclining, viewing TV). Sedentary behaviour was the sum of time reported on the 12 items whereas partial and absolute sedentary behaviours were the sum of their respective 6 items. Appendix A shows questions and items used to measure sedentary behaviour. Frailty was measured with the 15-item Tilburg Clinical Frailty Index (see Appendix B) with two descriptive anchors (yes – 1; no – 0) adopted in whole from a previous study [26]. As noted earlier, we used this clinical measure of frailty to be able to identify implications for clinical or geriatric practice. This scale produced a satisfactory internal consistency in the form of Cronbach’s alpha coefficient = 0.92.]

→How much has the validity of this measure of physical activity been proven in the past? I would appreciate it if you could tell me about previous studies. Also, I thought the study would have been more reliable and valid if, in addition to the questionnaire, an objective measure (such as using an activity meter) had been taken. Why did you not use an activity meter?

2.[The participants were community-dwelling older adults aged 50 years or higher in Ghana. We targeted older adults in two peri-urban towns, each with a combination of low and high socio-economic neighbourhoods. The following inclusion criteria were used to select the participants: (1) having at least a basic education qualification (i.e., basic school living certificate), which we used as an indicator of the ability to complete the questionnaire in English; (2) being a permanent community-dwelling resident aged 50 years or higher, and (3) the ability to walk independently for at least 10 minutes [24]. We calculated the minimum sample size necessary for this study with the G*Power software and recommended statistics (i.e., effect size = 0.2, α = 0.05, power = 0.8) suited for our sample [25]. The minimum sample estimated for HLR analysis with a maximum of 10 predictors was 91. To maximise the statistical power of our tests, we selected as many eligible individuals as possible. There was no sampling frame for this study, so potential participants were selected by research assistants at community and social centres such as supermarkets and events. Research assistants interviewed potential participants at various community centres to assess their eligibility. A total of 1039 older adults who met the above criteria were selected.]

→With regard to the eligibility of the subjects, since the study was conducted on elderly people, was any cognitive function testing done? I was concerned about the reliability of the study since it was mainly a questionnaire-based study.

Reviewer #2: This study uses a unique sedentary time questionnaire for the elderly and examines its correlation with frailty. Although this is an interesting study, the main concerns are listed below.

1. The introduction talks about the reliability of surveys using questionnaires for elderly people, and I think this point is important. Please add a little more detail about whether this can be compensated for by the 12-item questionnaire of this study. Furthermore, “they do not capture human

"feelings or psychology, are generally expensive, and can easily be damaged", but the meaning of this part is unclear.

2. This study interprets the results based on correlation only, but has the two groups, the frailty group and the robust group, been compared? I think it would be more persuasive if this data was also shown.

6. PLOS authors have the option to publish the peer review history of their article (what does this mean?). If published, this will include your full peer review and any attached files.

Reviewer #1: No

Reviewer #2: No

---

## [Decision Letter · Decision Letter 1]

13 Oct 2023

Associations of Frailty with Partial and Absolute Sedentary Behaviours among Older Adults: A STROBE-Compliant Analysis of Modifiability by Gender and Age

PONE-D-23-22789R1

Dear Dr. Asiamah,

We’re pleased to inform you that your manuscript has been judged scientifically suitable for publication and will be formally accepted for publication once it meets all outstanding technical requirements.

Kind regards,

Kiyoshi Sanada, PhD

Academic Editor

PLOS ONE

Additional Editor Comments (optional):

Reviewers' comments:

Reviewer's Responses to Questions

**Comments to the Author**

1. If the authors have adequately addressed your comments raised in a previous round of review and you feel that this manuscript is now acceptable for publication, you may indicate that here to bypass the “Comments to the Author” section, enter your conflict of interest statement in the “Confidential to Editor” section, and submit your "Accept" recommendation.

Reviewer #1: All comments have been addressed

Reviewer #2: All comments have been addressed

2. Is the manuscript technically sound, and do the data support the conclusions?

Reviewer #1: Yes

Reviewer #2: Yes

3. Has the statistical analysis been performed appropriately and rigorously? 

Reviewer #1: Yes

Reviewer #2: Yes

4. Have the authors made all data underlying the findings in their manuscript fully available?

Reviewer #1: Yes

Reviewer #2: Yes

5. Is the manuscript presented in an intelligible fashion and written in standard English?

Reviewer #1: Yes

Reviewer #2: Yes

6. Review Comments to the Author

Reviewer #1: Thank you very much for your precise response to my question. I understand it well now.

I will be able to apply the contents to my research in the future.

We look forward to your future activities. Thank you very much.

Reviewer #2: I fully understand the author's response. In the future, we hope to increase the number of subjects and conduct more advanced research.

7. PLOS authors have the option to publish the peer review history of their article (what does this mean?). If published, this will include your full peer review and any attached files.

Reviewer #1: No

Reviewer #2: No

---

## [Editor Report · Acceptance letter]

18 Oct 2023

PONE-D-23-22789R1 

Associations of Frailty with Partial and Absolute Sedentary Behaviours among Older Adults: A STROBE-Compliant Analysis of Modifiability by Gender and Age 

Dear Dr. Asiamah:

I'm pleased to inform you that your manuscript has been deemed suitable for publication in PLOS ONE. Congratulations! Your manuscript is now with our production department. 

Kind regards, 

on behalf of

Dr. Kiyoshi Sanada 

Academic Editor

PLOS ONE